# The Increased Mortality Rate with Lower Incidence of Traumatic Brain Injury during the COVID-19 Pandemic: A National Study

**DOI:** 10.3390/healthcare10101954

**Published:** 2022-10-06

**Authors:** Grzegorz Miękisiak, Dariusz Szarek, Samuel D. Pettersson, Celina Pezowicz, Piotr Morasiewicz, Łukasz Kubaszewski, Tomasz Szmuda

**Affiliations:** 1Institute of Medicine, University of Opole, 45-040 Opole, Poland; 2Department of Neurosurgery, Marciniak’s Hospital, 54-049 Wrocław, Poland; 3Department of Neurosurgery, Medical University of Gdansk, 80-210 Gdansk, Poland; 4Department of Mechanics, Materials and Biomedical Engineering, Faculty of Mechanical Engineering, Wrocław University of Science and Technology, Łukasiewicza 7/9, 50-371 Wrocław, Poland; 5Adult Spine Orthopaedics Department, Poznan University of Medical Sciences, 61-545 Poznan, Poland

**Keywords:** traumatic brain injury, head trauma, COVID-19, mortality, lockdown

## Abstract

Background: the COVID-19 pandemic with the following lockdown strategies have affected virtually all aspects of everyday life. Health services all over the world faced the crisis on an unprecedented scale, hampering timely care delivery. The present study was designed to assess the impact of the COVID-19 outbreak on the incidence and treatment of traumatic brain injuries in Poland. Methods: the data on hospital admissions with traumatic brain injuries as the primary diagnosis were extracted from the National Health Fund of Poland. For the purpose of this study, the search was limited to four relevant diagnosis-related groups. The overall in-house mortality was calculated. Results: there were 115,200 hospitalizations due to traumatic brain injury identified in the database. Overall, in comparison with the average of six prior years, in 2020 the volume of patients with traumatic brain injury dropped by 24.68% while the in-house mortality rate was increased by 26.75%. Conclusions: the COVID-19 pandemic with the resulting lockdown caused a radical reduction in human mobility. It had a profound impact on the incidence of traumatic brain injury, which dropped significantly. At the same time, the mortality rate increased drastically.

## 1. Introduction

Traumatic brain injury (TBI) is one of the leading causes of permanent disability worldwide [1]. It is sometimes referred to as a silent epidemic [2] and its incidence is still growing. Each year, an estimated 69 million individuals will suffer a TBI worldwide [3] with an estimated incidence of severe TBI of 73 cases per 100,000 people [4]. TBIs require multidisciplinary approaches and constitute a significant burden for all healthcare systems, regardless of socioeconomic status. The timely and effective prehospital as well as in-hospital treatment by qualified multidisciplinary teams is imperative to reduce mortality and morbidity [5]. This requirement makes it particularly reliant on efficient and robust emergency medical care, which was heavily affected by the recent COVID-19 pandemic.

It has already been shown that this unprecedented global outbreak profoundly impacted the management of neurosurgical patients. According to a recent study by Jean et al. [6], the volume of neurosurgical operations has dropped by more than 50% globally since the onset of the pandemic. As health services worldwide were inundated with infectious cases, most non-emergency procedures were deferred [7]. In March 2020, the American College of Surgeons issued prodigious recommendations to, among other things, “reschedule elective surgeries as necessary, and shift elective, urgent inpatient diagnostic and surgical procedures to outpatient settings, when feasible” [8].

The lockdowns have changed the behavior of the population on an unparalleled scale. This, in turn, affected the pattern of trauma, including TBI. It has been reported that lower mobility due to the pandemic resulted in a lower rate of road traffic accidents [9], but on the other hand, the incidence of violence-related TBIs increased [10].

This work aimed to assess the combined effect of the limited capacity of acute healthcare and altered patterns of trauma on the overall immediate outcomes of TBI treatment during the COVID-19 pandemic. This was accomplished by evaluating the national report on all patients treated for this particular type of injury in Poland for six consecutive years, from 2015 to 2020. The emphasis was on analyzing demographics, general volume, and in-house mortality. The main research question was whether the limited access to acute neurosurgical services was a more robust outcome factor than the purported decreased incidence of trauma overall.

## 2. Materials and Methods

### 2.1. Background

Poland is located in central and eastern Europe, with a population of 38,162,000 [11]. According to the World Bank, it is considered a high-income country [12]. The source of data for this study was the National Health Fund of Poland (Narodowy Fundusz Zdrowia—NFZ), which manages healthcare in Poland and is financed by compulsory health insurance contributions. In the last 14 years, NFZ has been using diagnosis-related groups for reimbursement. The pertaining data are regularly published on an official website [13]. The data used in this study covered all hospitalizations, with the TBI as a primary diagnosis funded by the National Health Fund within the period of 1 January 2015–31 December 2020. The latter year was representative of the pandemic lockdown. The following reimbursement groups were analyzed: A01—TBIs treated surgically and A76—TBIs managed conservatively. In 2018, two groups were added: PZA01 and P35, for minors under 18 treated respectively surgically or conservatively for TBI. The above groups cover all relevant hospitalizations. As TBIs are all treated in Poland within the public sector, the study covered all patients treated for TBI nationwide.

### 2.2. Study Design

This was a retrospective cohort study.

### 2.3. Participants

For the calculation of incidence, this work covers the entire population of Poland during the period specified above. The inclusion criterion was reimbursed hospitalization within A01, A76, PZA01 and P35 groups. There were no exclusion criteria. The sample size calculations for the population of 12,849 in 2020 revealed an error of 0.01% for the mortality rate.

### 2.4. Outcome Measures

In-house mortality, incidence.

### 2.5. Statistical Analysis

The significance of the changes in hospital admissions were calculated using the unadjusted Poisson regression model. The incidences were compared by calculating the z-ratio for the significance of the difference between two independent proportions [14,15]. Confidence intervals for proportions were calculated using the Wilson procedure without a correction for continuity [15]. The value of *p* < 0.05 was considered statistically significant.

The software used for statistical analyses were StatsDirect version 3.3.5 (StatsDirect Ltd., Merseyside, UK; http://www.statsdirect.com) and MedCalc v. 12.5.0.0 (MedCalc Software, Ostend, Belgium).

## 3. Results

During 2015–2020, a total of 115,200 hospitalizations were recorded. The overall number of TBI hospitalizations decreased in 2020 for both conservative and surgical management; the difference was statistically significant (Figure 1). In regard to diagnoses, significant differences were noted in the “surgical” group: there were fewer patients with subdural hematomas in favor of focal injuries and epidural hematomas (Table 1). Likewise, in the “conservative” group, a shift towards more severe conditions was noticeable (also Table 1). The breakdown of surgical procedures performed shows only minimal changes during 2020 against earlier years (Table 2). Although the age profile of patients was virtually unchanged in the “surgical” group, in a “conservative” group, the patient population was significantly older at *p* < 0.001 (Table 3). The most crucial finding was a statistically significant (*p* < 0.001) increase in the in-house mortality rate (Table 4). In either “surgical” and “conservative” groups, this value was increased in 2020, with, however, nearly identical total numbers of in-house deaths when comparing a sum from 2020 and mean per-annum from previous years. Overall, in comparison with the average of 6 prior years, in 2020 the volume of TBI patients dropped by 24.68% while the in-house mortality rate was increased by 26.75%.

## 4. Discussion

Poland was hit heavily by the recent COVID-19 pandemic, similarly to other countries in the region. On 4 March 2020 the first case of SARS-CoV-2 was registered and soon after, restrictions were introduced abruptly. On 12 March the government declared a national state of emergency, and three days later, the borders were closed. Lockdown rules were imposed and all citizens were required to stay at home unless necessary. On 20 March the authorities announced the introduction of an epidemic state in Poland. The restrictions were gradually raised beginning on 20 April. The second wave appeared in October and was much more severe than the first one. New rules were introduced again and remained in effect until the end of the year. According to official sources, there were 1,294,878 cases of SARS-CoV-2, with 28,554 deaths confirmed across the country [16]. The health service system was working near its maximum capacity from mid-March throughout the rest of that year.

The present study was designed to look at the national data, covering all TBI hospitalizations in the country during 2020 and 6 years prior. It revealed that the volume of TBI patients dropped significantly, by nearly 25%, during the COVID-19 pandemic. Other authors around the world have reported similar trends. A large study based on data from 85 trauma centers in the USA has shown that the early phases of the COVID-19 pandemic were associated with a 32.5% decrease in trauma patient volumes [17]. Another two articles from Austria [18] and Finland [19] have shown a significant reduction in the incidence of TBIs. The meta-analysis, totaling 18,490 patients from 13 studies, came to the same conclusion [10]. The primary rationale for the finding is believed to be the lockdown and the stay-at-home strategies during the pandemic, which significantly reduced foot and vehicle traffic [20,21,22,23], as the leading cause of TBI in Poland [24] is motor vehicle accidents. In a recent article by Johnson et al. [25] the authors further dissected the data on the incidence of TBIs during the pandemic. They found that the 18% decrease in the rate of TBI-related admissions was noted only during the first eight weeks of the lockdown, followed by a 16% increase during the last eight weeks. Cumulatively, there was no difference between the pre-pandemic and pandemic periods.

The epidemiology of other neurologic conditions, such as stroke, were also affected. During the COVID-19 period, the overall number of hospitalizations due to stroke in Poland dropped by 8.28% compared with the corresponding period of 2019 [26]. Other authors reported similar findings, with the highest drop of 39.52% during the critical period recorded in Spain [27]. The primary factor behind these findings is poor access to emergency medical services and fear of contracting the virus rather than changes in incidence [28]. The lockdown due to COVID-19 pandemic greatly restricted the mobility of entire populations, which turned into a significant decrease in the incidence of trauma worldwide [29], including the pediatric population [30]. 

The patient demographics changed during 2020. The patient population was significantly older, but only in the “conservative” group. This may be related to the change in the mechanisms of injury: the second most common cause of TBI is falls [24] and they primarily affect the elderly, the leading cause of death for persons 65 years of age or older [31,32]. The most prevalent diagnosis in both groups was subdural hemorrhage, with a small but significant increase in incidence in 2020. A similar finding was reported by Lara-Reyna [33]. On the contrary, a study from the Czech Republic and Austria showed no difference in all subgroups of patients treated for TBIs [34].

The COVID-19 pandemic profoundly affected the entire health system, and neurosurgery was not exempt. The survey study among neurosurgeons around the world by Jean et al. [6] has shown that nearly half of the respondents (46.1%) reported that their operative volume had dropped more than 50% during the pandemic. The drop was caused primarily by the postponement of all non-urgent operations that did not require urgent or emergent intervention [21]. However, patients with common time-sensitive medical and surgical emergencies, such as TBI, were to receive the necessary treatment without delay. Official recommendations were [35] established [8] to maintain the emergency care system. The authors of two studies [23,34] concerned with the non-elective neurosurgical procedures in central and eastern Europe have shown no difference in volume during the pandemic.

The key finding of this study is a significant increase in the mortality rate. In 2020 this value increased by nearly one-third, from 10.78% to 13.67%. This finding was replicated in two studies from India [36,37] but not in articles from Germany [38] or Ireland [39]. Most notably, the recent meta-analysis [10] has shown that mortality was increased only in low- to middle-income countries. As a high-income country, Poland may be a disturbing exception to this rule.

The treatment of TBI is immensely complex, and many factors affect the overall outcome. The available dataset does not provide a clear rationale for this finding. One plausible explanation for the increased mortality is the critical shortage of acute beds in ICUs [40], which were overburdened with COVID-19 patients requiring prolonged ventilation. Another problem of TBI care during pandemics was noticed by Manivannan et al., who found that a more significant proportion of patients are being managed locally in non-neurosurgical units in order to minimize the risk of COVID-19 spread between hospitals [41]. Regardless of the rationale, some 300 surplus in-house deaths were due to TBI in 2020 alone. As the total number of deaths did not change from previous years, it does not contribute to the widely reported [42] excessive all-causes mortality during pandemic.

## 5. Study Limitations

The present study is not without limitations. There were no available data on the mechanisms of TBI, which must have had a significant and demonstrable bearing on the outcomes. The severity of injury, a key prognostic factor, was also unobtainable. Mortality was the only reliable outcome measure given the available dataset, as information on discharge was very crude, not allowing for meaningful analysis. There was no available information on comorbidities nor concurrent drug therapies, which are both risk factors for mortality in COVID-19 patients in previous studies [43]. Last but not least, the data were pooled, and only predetermined statistics were available.

## 6. Conclusions

The COVID-19 pandemic and the subsequent lockdown caused an unparalleled reduction in human mobility. It had a profound impact on the incidence of TBI, which decreased by nearly 25% during COVID-19, while at the same time, the mortality rate increased by 26.75%, unlike in other high-income countries. The latter finding shows the necessity to change health policies during similar crises in the future.

## Figures and Tables

**Figure 1 healthcare-10-01954-f001:**
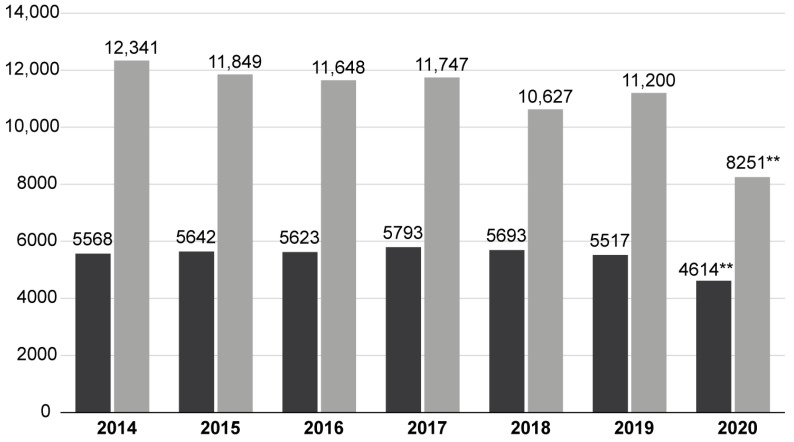
Comparison of the annual TBI patient volume between the pre-COVID-19 period and 2020. ** *p* < 0.001.

**Table 1 healthcare-10-01954-t001:** Diagnoses of hospitalized patients during the COVID-19 pandemic and preceding years according to the ICD 10 classification. * *p* < 0.05, ** *p* < 0.001.

Diagnosis	ICD 10	Surgical				Conservative			
		2020		2014–2019		2020		2014–2019	
		*n*	%	*n*	%	*n*	%	*n*	%
Diffuse brain injury	S06.2	143	3.11	1109	3.28	0	0	0	0
Focal brain injury	S06.3	87	1.89 *	837	2.47 *	1511	18.31 *	11,897	17.36 *
Epidural haemorrhage	S06.4	443	9.64 *	3643	10.77 *	257	3.11	2093	3.05
Traumatic subdural haemorrhage	S06.5	3639	79.16 **	26.113	77.17 **	1610	19.51 **	10,241	14.95 **
Traumatic subarahnoid hemorrhage	S06.6	0	0	0	0	685	8.3 **	4011	5.85 **
Cranial vault fracture	S02.0	0	0	0	0	1464	17.74 **	7667	11.19 **
Skull base fracture	S02.1	0	0	0	0	271	3.28 **	3181	4.64 **
Complex cranial fracture	S02.7	0	0	0	0	374	4.53 *	3535	5.16 *
Other	-	285	6.2	2136	6.31	2079	25.2 **	25,889	37.79 **
**total**		**4597**	**100**	**33,838**	**100**	**8251**	**100**	**68,514**	**100**

**Table 2 healthcare-10-01954-t002:** Main surgical procedures during the COVID-19 pandemic and preceding years according to the ICD 9 classification.

Surgical Procedure		2020		2014–2019	
		*n*	%	*n*	%
Burr hole	01.243 01.092	1728	34.26	12,222	34.15
Craniotomy for epidural hematoma	01.245	545	10.8	4115	11.5
Craniotomy for subdural hematoma	01.247	2510	49.76	17,771	49.66
Other	-	261	5.17	1677	4.69
**total**		**5044**	**100**	**35,785**	100

**Table 3 healthcare-10-01954-t003:** Age of patients treated due to TBI between the pre-COVID-19 period and 2020. * *p* < 0.05, ** *p* < 0.001.

Age Category	Surgical				Conservative			
	2020		2014–2019		2020		2014–2019	
	*n*	%	*n*	%	*n*	%	*n*	%
under 18	84	1.83	729	2.15	1044	12.65 **	11,736	17.13 **
18–40	582	12.66	4640	13.71	1643	19.91 **	16,128	23.54 **
41–60	1255	27.3	9218	27.25	2148	26.03 *	17,036	24.86 *
61–80	1808	39.33 *	12.811	37.87 *	2252	27.29 **	16,073	23.46 **
80+	868	18.88	6435	19.02	1164	14.11 **	7541	11.01 **
**total**	**4597**	**100**	**33,833**	**100**	**8251**	**100**	**68,514**	**100**

**Table 4 healthcare-10-01954-t004:** Comparison of mortality between the pre-COVID-19 period and 2020. ** *p* < 0.001.

	Surgical		Conservative		All	
	2020	2014–2019	2020	2014–2019	2020	2014–2019
N of deaths	1196	7675	560	3361	1756	11,036
Deaths per annum	1196	1279.17	560	560.17	1756	1839.34
N of hospitalizations	4597	33,833	8251	68,514	12,848	102,347
**Mortality rate (CI95%) [%]**	**26.02 (24.77–27.31) ****	**22.68 (22.24–23.13) ****	**6.79 (6.27–7.35) ****	**4.91 (4.75–5.07) ****	**13.68 (13.09–14.27) ****	**10.78 (10.59–10.97) ****

## Data Availability

Original data were obtained from the official website of the NFZ, which is available in the public domain.

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
