# Peer review of "The Increased Mortality Rate with Lower Incidence of Traumatic Brain Injury during the COVID-19 Pandemic: A National Study"

_healthcare, 2022, doi:10.3390/healthcare10101954_

Round 1
Reviewer 1 Report
Thank you for the opportunity to review your work. In my opinion, the article deviates from MDPI standards.
- First of all, all of its sections are too short - the two paragraphs of the introduction do not give the reader the opportunity to get acquainted with the topic, moreover, they do not indicate the research gap that the Authors want to fill.
- The purpose of the paper is also unclear; the Authors did not state hypotheses or research questions.
- The methodology is described in a haphazard manner. It is suggested that the description of the material and methods should designate thematic sections, e.g.: study area, participants, inclusion criteria, research tools, statistical analyses, etc.
- The authors suggest that chi-square statistics were used, but nowhere are these statistics reported. Tables should report the results of the inference in the form of test value, strength/direction of relationship index and p-value.
- The discussion is also inadequate. Less than 15 sources were used, where the topic of COVID-19 is widely discussed in the scientific world. As of the date of the review, I see 707 records in the PubMed database - so the material for the search exists.
- At the end of the discussion, the authors did not bend to the limitations of their research.
I request that the paper be rejected and that consideration be given to possibly resubmitting it after significant improvements.
Author Response
Dear Reviewer,
Thank you for your comprehensive review. I hope we were able to address all issues.
The manuscript underwent a thorough upgrade. All sections have been elongated, the introduction in particular. The methods were revamped and the instead of chi-square another method was adopted for this study. We added more resources to the discussion section. One particular metaanalysis published last month has made our conclusions even more intriguing. Last but not least we added the paragraph on limitations.
Sincerely,
On behalf of all authors
- Miekisiak
Reviewer 2 Report
Dear authors,
The article they present is interesting.
It evidences the changes in clinical care brought about by the pandemic.
However, improvements are needed.
Introduction
It is insufficient.
It is necessary to introduce the study correctly.
I propose to include national and global TBI incidence and prevalence data; data on the process of admission, admission and rehabilitation of TBI in the Polish healthcare system; and national and international mortality and comorbidity data.
In addition, we need to know what health care has been like in Poland for TBI during the pandemic.
Methods
It is necessary to know how the sample was accessed, sample size calculations, research team, statistical treatment of the data.
Discussion
There is a need to increase the comparison and discussion of international and global data.
Kind regards,
Author Response
Dear Reviewer,
Thank you for your comprehensive review. I hope we were able to address all issues.
The manuscript underwent a thorough upgrade. All sections have been elongated, the introduction in particular. The methods were revamped and the instead of chi-square another method was adopted for this study. We added more resources to the discussion section. One particular metaanalysis published last month has made our conclusions even more intriguing. Last but not least we added the paragraph on limitations.
Unfortunately we were unable to provide more detailed data on comorbidities nor the admission / discharge information. Same applies to information on rehabilitation after TBIs: we were constrained to the analyses published by the insurer.
Sincerely,
On behalf of all authors
G. Miekisiak
Round 2
Reviewer 1 Report
The authors have made the required changes. Please accept the publication in its current form.
Author Response
Dear Reviewer,
Thank you for your effort, your quick and thorough review is much appreciated
Sincerely,
on behalf of all authors,
G. Miekisiak
Reviewer 2 Report
Dear authors,
The article includes improvements. However, further changes are needed to make it publishable.
Abstract:
Do not use abbreviations.
Methods:
The use of abbreviations should be avoided. They are complex to read.
I suggest dividing into:
Study design
Participants
Outcome measures
Statistical analysis
Results:
Include legends in the tables.
To elaborate small tables that allow a quick and understandable reading.
Comment in the text the most important results of each table or figure.
Kind regards,
Author Response
Dear Reviewer,
Thank you for your effort, I much appreciate your instant review.
I addressed all issues you raised except for the main acronym in the manuscript: TBI. We have tried to replace it, but the clarity of the text was significantly compromised, and thus we decided to leave this acronym as is.
The methods section was divided into subsections, as you requested.
The large table was divided into three, and the results section was altered accordingly.
Sincerely,
On behalf of all authors,
G. Miekisiak